# Elastic Electrically Conductive Composites Based on Vapor-Grown Carbon Fibers for Use in Sensors

**DOI:** 10.3390/polym15092005

**Published:** 2023-04-24

**Authors:** Ahmed Nasr, Ondřej Mrhálek, Petr Svoboda

**Affiliations:** Department of Polymer Engineering, Faculty of Technology, Tomas Bata University in Zlin, Vavreckova 5669, 760 01 Zlin, Czech Republic

**Keywords:** carbon fibres, ethylene-octene copolymer, electrical properties, mechanical properties

## Abstract

Elastic electrically conductive composites with an ethylene octene copolymer matrix (EOC) and vapor-grown carbon fibers (VGCF) were prepared by ultrasonication in a toluene solution, and their morphology, mechanical and electrical properties were also evaluated. EOC/CF composites were estimated for their mechanical and viscoelastic properties. The morphology of the composites was analyzed using scanning electron microscopy (SEM), and stress–strain curves were generated to measure the stress and tensile modulus of the composites. The experimental results were compared with various theoretical models, including the Burgers model, which separates viscoelastic behavior into several components. A dynamic mechanical analysis was also used to measure the composites’ storage modulus, loss modulus, and damping factor at different frequencies. The composites’ complex viscosity and storage modulus were increased with higher wt.% of CF, which enhances the elastic response. Electrical resistivity measurements were conducted on the composites and it was found that the resistivity decreased as the sample was loaded and increased as it was unloaded. Overall, the study provides insights into the mechanical and viscoelastic properties of EOC/CF composites, which could be helpful in developing sensors such as pressure/strain sensors.

## 1. Introduction

Nanotechnology has recently become an increasingly popular area of research, with a particular interest in using nanofillers in composite materials. One area of focus for researchers has been the development of conductive polymer composites (CPCs) for use in various electronic applications, including pressure–strain sensors, organic vapor detectors, actuators and temperature sensors [1,2,3,4]. Carbon fibers (CFs) have long been considered an excellent conductive filler for these composites, with research into their use dating back to 1960. In particular, vapor-grown carbon fibers (VGCFs), which were first developed in 1980, have gained popularity as a reinforcement material for composite materials due to their high strength and modulus [5,6].

Percolation theory can be used to explain the mechanism of electrical conductivity in CPCs, where the filler creates a continuous conductive pathway. The percolation threshold refers to the minimum volume fraction of the filler required to make this pathway. Elastic composites may experience a significant decrease in electrical conductivity when stretched due to the disruption of the conductive pathways [2,7]. The polymer matrix in these composites can be various [8,9]. Nevertheless, thermoplastic elastomers have grown in recent years due to their unique properties, which combine processability as thermoplastics and rubbery properties as elastomers. They are a particular class of polymers that generally consist of hard and soft phases [10]. Lozano-Perez et al. conducted a study to examine the impact of the rigid segment on electromechanical behavior. The findings demonstrated that an increased content of rigid segments increases the tensile modulus, mechanical hysteresis, and electrical response [11].

Several researchers have investigated the mechanical properties of CPCs [12]. Zhang et al. studied the strain–sensing behavior of elastomeric composites under mechanical cyclic loading [13]. The electrical resistance of the elastomeric composites shows perfect recoverability after cyclic loading. The conductivity and electrical resistance depend on the shape of the filler particles. A comparative study with various carbon fillers was investigated by Theravalappil et al. They found that the longer carbon fibers create conductive paths at lower concentrations than multi-walled carbon nanotubes [14]. Even though using thermoplastic elastomers as a matrix for CPCs has already been investigated, little attention has been paid to copolymers of polyethylene/poly(α-olefin), e.g., ethylene-octene copolymer (EOC).

This study focused on the preparation of elastic CPCs using EOC and CF and the subsequent evaluation of their mechanical and electrical properties. The resulting EOC/CF composites demonstrated exceptional elasticity, making them highly suitable for use in pressure and strain sensors.

## 2. Experimental

### 2.1. Materials

The Dow Chemical Company supplied the ethylene-octene copolymer Engage 8842 used in this study. The octene content was 45 wt.%, the density 0.857 g/cm^3^ and the melt flow index 1.0 g/10 min (190 °C/2.16 kg). Carbon fibers were provided by Showa Denko, Japan, under the trade name VGCF (vapor-grown carbon fibers). The properties of VGCF are represented in Table 1.

### 2.2. Sample Preparation and Morphology

An EOC matrix with 10, 15, 20, 25, 30 and 35 wt.% of fillers was blended by ultrasonication method. First, the calculated amount of filler was added to the EOC/toluene solution.The sonication process was then applied at 80 °C (Dr. Hielscher GmbH apparatus; amplitude 88 µm, power density 300 W/cm^2^, and frequency 24 kHz) for one hour. The composite was precipitated with acetone and dried at 50 °C for 12 h. Finally, the samples were prepared by compression molding at 100 °C in the hydraulic press and cut into a rectangular shape (50 × 10 × 0.5 mm). The morphology and dispersion of CF in the polymer matrix were analyzed by scanning electron microscopy (SEM) on a Vega LMU Tescan with a voltage of 10 kV. Before analysis, samples were put into liquid nitrogen and were broken after 1 min.

### 2.3. Mechanical Properties

Tensile stress-strain measurement, creep behavior and frequency sweep were carried out using a Mettler Toledo DMA1 at room temperature. The stress–strain experiment was set up from 0 to 5 N with the force changing at a rate of 0.5 N/min. From the beginning of the stress–strain curves, the tensile modulus was evaluated. Creep behavior was tested at standard tensile mode with various loads for 5 min. Force 0.2, 0.5, 1, 2, 3, 4 and 5 N was applied to connect mechanical with electrical properties. At the frequency sweep measurement, frequencies were set up from 0.1 to 100 Hz. Storage modulus (E′) and Tan δ were evaluated.

### 2.4. Electrical Properties

The samples were put between the clamps of a multimeter UNI-T UT71C and electrical resistance was measured. Various calibrated weights (20, 50, 100, 200, 300, 400 and 500 g) were then placed on the lower clamp holding the sample and the electrical resistance was measured again after stretching. First, the sample was loaded with the weight for 5 min; then, it was unloaded for 5 min. This process was repeated. The test was continuous, and the electrical resistance was recorded every 1 s by software UNI-T UT71ABC. The measurements were performed at room temperature (25 °C).

## 3. Results and Discussion

### 3.1. Sample Morphology

The morphology of EOC/CF composites was obtained using scanning electron microscopy (SEM). Figure 1 shows the morphology of the composite with 30 wt.% CF. As can be seen, filler particles are uniformly dispersed, proving the efficiency of ultrasonication mixing. The conductivity path exists in a place where the fibers are crossed. In fact, there are more contacts in the sample volume (invisible on SEM). The picture shows only the sample’s surface after the fracture break at low temperatures.

### 3.2. Mechanical Properties

Stress–strain curves are illustrated in Figure 2. Stress increases with a higher content of the filler for the same strain, implying a higher modulus. The reinforcing effect of the CF filler in the EOC matrix causes this. Even though the stress and tensile modulus are higher, EOC/CF composites remained elastic.

Several theoretical models have been described in the literature to elucidate the mechanical properties of composites containing different types of fillers. Among these models, the hydrodynamic theory proposed by Einstein [15] is considered the earliest one for spherical filler particles, which explains the viscosity of colloidal suspensions [15].
(1)$$\eta =\eta_{0}\;\left(1+2.5\;\phi \right)\;$$
where *η* is the viscosity of the suspension, *η*_0_ is the viscosity of the incompressible fluid and *ϕ* is the volume fraction of the spherical particles.

This model predicts an increase in viscosity due to the presence of the filler. Einstein’s theory also assumes that spherical particles are uniformly dispersed in the polymer matrix and perfectly bonded with the matrix. Guth and Gold generalize Einstein’s equation to predict the tensile modulus of filled composites instead of the viscosity. Furthermore, this model includes and explains the interaction between the matrix and fillers at a higher content of filler [15,16,17,18]. The Guth–Gold model is shown in Equation (2):(2)$$\frac{E_{c}}{E_{0}}=\left(1+2.5\;\phi +14.1\;\phi^{2}\right)$$
where *E_c_* is the tensile modulus of the composite, *E*_0_ is the unfilled polymer matrix’s tensile modulus and *ϕ* is the filler’s volume fraction. Equation (2) describes the increase in the tensile modulus of the composite with spherical particles as a function of the filler content. Nevertheless, it was found that in the case of non-spherical particles, the tensile modulus could increase more than is predicted by Equation (2) [15,17]. Considering the shape of the fillers, Guth and Smallwood presented the shape factor f for non-spherical particles in Equation (3) [19]:(3)$$\frac{E_{c}}{E_{0}}=\left(1+0.67\;f\phi +1.62\;f^{2}\phi^{2}\right)$$

The results presented in Figure 3 compare the experimental tensile modulus of EOC/CF composites with the Guth–Gold model for spherical particles and the Guth–Smallwood model for non-spherical particles, with varying shape factors. As the weight percentage of CF increases from 0% to 30%, the tensile modulus of the composites significantly increases from 4 MPa to 28 MPa, respectively. The obtained experimental data agree with the Guth–Smallwood model for non-spherical particles, which suggests that the carbon fibers have a non-spherical shape with a shape factor of 0.5 [20].

Several models describe and evaluate the creep behavior of polymer materials, exhibiting both elastic and viscous responses when a force is applied. Viscoelastic parameters from these models can be used to predict polymer creep deformation mechanisms. One of these models is the four-parameter model, known as the Burgers model, which can separate viscoelastic behavior into several components: an instantaneous elastic response, a retarded elastic response and a viscous response (Figure 4) [21,22,23].

The Burgers model consists of the Maxwell and Kelvin models connected in series. The following equations determine the overall creep strain of the Burgers model:(4)$$\varepsilon =\varepsilon_{1}+\varepsilon_{2}+\varepsilon_{3}$$
(5)$$\varepsilon =\frac{\sigma_{0}}{E_{M}}+\frac{\sigma_{0}}{E_{K}}\left(1-e^{\frac{-E_{K}t}{\eta_{K}}}\right)+\frac{\sigma_{0}}{\eta_{M}}t$$
where *ε*_1_ is the strain of the Maxwell spring, *ε*_2_ is the strain of the Kelvin unit and *ε*_3_ is the strain of the Maxwell dashpot. *E_M_* and *η_M_* are the modulus and viscosity of the Maxwell spring and dashpot. *E_K_* and *η_K_* are the modulus and viscosity of the Kelvin spring and dashpot. In addition, σ_0_ and t are the applied stress and creep test time, respectively.

Furthermore, for another evaluation of creep behavior, we can define creep compliance *J*(*t*) as a ratio of the strain per unit of the applied stress according to the following equations [24,25,26].
(6)$$J\left(t\right)=\frac{\varepsilon \left(t\right)}{\sigma_{0}}$$
(7)$$J\left(t\right)=\frac{1}{E_{M}}+\frac{1}{E_{K}}\left(1-e^{\frac{-E_{K}t}{\eta_{K}}}\right)+\frac{t}{\eta_{M}}$$

Nevertheless, this study also uses the six-parameter model, which combines the Maxwell model and two Kelvin models connected in series (Figure 4) [22]. These equations can define the total creep strain and respective creep compliance:(8)$$\varepsilon =\frac{\sigma_{0}}{E_{0}}+\frac{\sigma_{0}}{E_{1}}\left(1-e^{\frac{-E_{1}t}{\eta_{1}}}\right)+\frac{\sigma_{0}}{E_{2}}\left(1-e^{\frac{-E_{2}t}{\eta_{2}}}\right)+\frac{\sigma_{0}}{\eta_{0}}t$$
(9)$$J\left(t\right)=\frac{1}{E_{0}}+\frac{1}{E_{1}}\left(1-e^{\frac{-E_{1}t}{\eta_{1}}}\right)+\frac{1}{E_{2}}\left(1-e^{\frac{-E_{2}t}{\eta_{2}}}\right)+\frac{t}{\eta_{0}}$$

According to Figure 5, the six-parameter model fitted the experimental data better than the four-parameter model. The parameter values of both models are included in Table 2 and Table 3.

In Figure 6, the creep compliance *J*(*t*) is depicted, which is a measure of the ability of a material to deform over time under constant stress. After 300 s from the start, the graph illustrates that the creep compliance decreases by increasing the weight percentage of CF in EOC composites. This indicates that EOC/CF composites with a higher weight percentage of CF exhibit less deformation over time under constant stress.

Figure 7a shows parameter *J*_0_ which represents 1/*E*_0_ from the six-parameter model. *J*_0_ decreases with a higher content of CF from 0.15 MPa^−1^ to 0.03 MPa^−1^. This indicates that the stiffness of the composite decreases as the amount of CF increases. This behavior is attributed to the fact that carbon fibers are more rigid than the polymer matrix and the decrease in stiffness is due to the reduction in the composite’s polymer amount.

Figure 7b shows the parameter *E_M_* from the Burgers model, which *E_M_* increases with an increasing wt.% of CF from 6.7 MPa to 31.7 MPa. The increase in *E_M_* with the increasing filler content is attributed to the forming of a percolated particle network structure, improving the composites’ mechanical and electrical properties [27].

The viscoelastic behavior of the samples was studied using dynamic mechanical analysis. This technique allowed for the measurement of crucial dynamic parameters such as storage modulus (E′), loss modulus (E′′), and damping factor (Tan δ) as a function of frequency. Typically, at a specific temperature, the storage modulus increases with a higher frequency [28,29].

As shown in Figure 8a, the storage modulus of EOC/CF composites increases with increasing frequency. The growth of the exponential slope is observed due to the various CF contents, ranging from 0.68 to 3.8 with an increasing wt.% CF. The equations and parameters that describe this relationship are included in Figure 8b. The increase in storage modulus with increasing frequency and filler content is a crucial factor in determining the stiffness and elastic response of the composites. This behavior is attributed to the forming of a percolated particle network structure, which improves the composites’ mechanical and electrical properties.

In elastic electrically conductive composites, an increase in the storage modulus with increasing filler content leads to an increase in complex viscosity, improving the elastic response. This is because the storage modulus is closely related to the material’s elasticity. It has been observed that as the filler content increases, there is a significant increase in the storage modulus of polymer composites, which leads to the formation of a percolated particle network structure [30,31]. In their study, Fernandez et al. noted that the frequency dependence of the storage modulus was reduced as the carbon fiber content increased, resulting in less viscoelastic behavior in the composites [32]. This observation highlights the filler content’s importance in determining the composites’ mechanical behavior. In Equation (10), Tan δ is defined as a ratio of the loss modulus and storage modulus:(10)$$\mathrm{Tan}\left(\delta \right)=\frac{E^{\prime \prime }}{E^{\prime }}$$

Viscoelastic liquids behavior is generally observed when Tan δ decreases with the frequency. On the other hand, the positive slope of Tan δ curves indicates elastic behavior [33]. Conversely, materials displaying an increase in Tan δ with frequency demonstrate their elastic behavior. Figure 9a shows the Tan δ behavior of elastic electrically conductive composites (EOC) with varying CF content, measured at room temperature. At lower CF content, Tan *δ* increases with increasing frequency, while no significant increase is observed for samples with 25 and 30 wt.% CF. Figure 9b illustrates the dependence of Tan δ on the CF content at a frequency of 0.1 Hz and includes an equation and parameters to describe the dependence.

### 3.3. Electrical Properties

Previous research by Theravalappil et al. has shown that elastic electrically conductive composites reinforced with carbon fibers (EOC/CF) have a percolation threshold at 10 wt.% CF. This result is lower than that observed for multi-walled carbon nanotube (MWCNT) composites due to the longer length of the carbon fibers, which allows for the formation of a conductive path at a lower concentration [14]. Generally, composites with filler concentrations close to the percolation threshold exhibit a high electrical resistance and gauge factor [9], making it difficult to measure their electrical properties accurately.

This study used EOC/CF composites with concentrations of 15, 20, and 25 wt.% CF was used, which is above the percolation threshold, allowing for the observation and analysis of their electrical behavior under strain. In our previous paper, we focused on AC conductivity [14] while this work is focused on DC conductivity change with stretching. The two loading and unloading cycles for the EOC composite with 25 wt.% CF and calibrated weight 50 g are presented in Figure 10. One cycle starts by loading the sample and electrical resistivity was measured every 1 s for 5 min. Measurement continues for 5 min with the unloading sample and the cycle ends. The sample loading process corresponds with the stress increase (Figure 10a) and decrease of the electrical resistivity and resistance change, respectively (Figure 10b,c). Furthermore, the unloading sample causes an increase in the electrical resistivity and resistance change.

In general, positive piezoresistivity, when the applied strain changes the electrical resistivity, is more usual than negative piezoresistivity. Many researchers reported results with positive piezoresistivity when the electrical resistance was increased with an applied tensile strain [34,35,36,37]. Nevertheless, Figure 10b shows that the electrical resistivity decreases with applied strain, which the realignment of carbon fibers could explain during the loading process. The schema of this process is illustrated in Figure 11. While loading, the sample is stretched and the length of the sample increases. The distance between two fibers also increases (a_2_ > a_1_), increasing the electrical resistivity. For the electrons, it is not very easy to find the conductive path. On the other hand, the width and thickness are reduced during sample loading, which causes a decrease in the distance between two fibers (b_2_ < b_1_) and creates new conductive paths. This effect supports the tunneling, which requires a small distance (order in Angstroms) of the adjacent fibers. The tunneling effect leads to a decrease in the electrical resistance. The overall behavior results from the competition between these effects [9,35,38].

The previous explanation can be applied to the negative resistance change (Figure 12) and negative gauge factor (Figure 13), which are defined in Equation (11):(11)$$GF=\frac{\Delta R/R_{0}}{\Delta l/l_{0}}$$
where $\Delta R/R_{0}$ is the relative resistance change, Δl is the change of the length during loading and $l_{0}$ is the length of the sample before loading.

Generally, most materials exhibit a positive gauge factor; their electrical resistance increases under tensile strain. However, some materials display a negative gauge factor, such as nickel (−12), n-type silicon (−135), and Si nanowires (−285). It is worth noting that negative gauge factors are relatively rare and not well understood. In contrast, polymer composites filled with semiconducting particles typically exhibit a positive gauge factor, indicating that the electrical resistance increases with tensile strain [39]. These observations are essential because they provide insights into the behavior of various materials under strain and can inform the selection and development of materials for sensor applications. Further work is needed to understand the underlying mechanisms responsible for the negative gauge factor behavior and to identify ways to mitigate it.

The strain dependence of resistance change for elastic electrically conductive (EOC) composites with varying weight percentages of CF (15, 20, and 25 wt.%) is depicted in Figure 12 and Table 4. The graph depicts the relationship between the strain, tensile stress, and calibrated weights ranging from 20 to 500 g. Unexpectedly, the resistance change decreases to negative values. This observation is significant as it suggests that the composite material exhibits non-linear behavior under stress, and its electrical properties may not be suitable for specific sensor applications.

Additionally, Figure 13 and Table 5 illustrate the fact that the gauge factor, which is a measure of a material’s sensitivity to strain, reaches negative values. The data presented in Table 5 show the linear regression parameters (y_0_, a, b, and R^2^) for each weight percentage of carbon fiber. Interestingly, the gauge factor for the 25 wt.% of composite is significantly lower than the gauge factors for the 15 wt.% and 20 wt.% composites and it also exhibits a negative y-intercept. These results suggest that the composite material’s electrical conductivity and strain sensitivity are strongly influenced by the weight percentage of carbon fiber [40,41]. The negative gauge factor values observed in this study are worrying as they indicate that the material’s electrical resistance decreases under tensile strain, which could lead to inaccurate sensor readings. Further investigation is needed to understand the underlying causes of this behavior and to develop strategies to improve the stability and reliability of EOC composites for use in sensor applications.

As can be seen in Figure 14, resistivity variations were observed during multiple loading and unloading cycles at a tensile stress of 1.202 MPa. The results indicate that the loading/unloading process is consistent across two cycles and a prolonged period, with no substantial changes in resistivity. Furthermore, the resistivity levels oscillate between two values, indicating the stability of the material. These findings highlight the importance of repeatability in pressure and strain sensors, as it ensures consistency in measurement accuracy over time [42]. Therefore, the ability of a material to maintain stable resistivity values during loading/unloading cycles is a critical factor to consider when selecting materials for use in pressure and strain sensors. The findings of this study suggest that carbon fiber-based composites may be suitable for use in these applications due to their ability to maintain stable resistivity values over prolonged periods.

## 4. Conclusions

In conclusion, the study followed the effect of carbon fibers in an elastic polymer matrix on the mechanical properties and morphology of EOC/CF composites, which were investigated using numerous analytical techniques. The SEM images confirmed the efficient dispersion of the CF filler in the EOC matrix, and the stress−strain curves showed that the addition of CF improved the tensile modulus and stress of the composites without sacrificing their elasticity. The mechanical behavior of the composites was also evaluated using theoretical models such as the Guth−Gold and Guth−Smallwood models, which were used to estimate the tensile modulus of the composites with different filler shapes. The viscoelastic behavior of the composites was evaluated using the Burgers model, and the dynamic mechanical analysis revealed that the storage modulus increased with frequency, and the Tan δ curves indicated elastic behavior. The electrical properties of the composites were also investigated, and the results showed that the EOC/CF composites exhibited a percolation threshold at 10 wt.% CF. Overall, this investigation points to unique combinations of thermoplastic elastomers and carbon fibers, which have implications for developing advanced composites with improved properties and can be used in electronics engineering, especially the pressure/strain sensors.

## Figures and Tables

**Figure 1 polymers-15-02005-f001:**
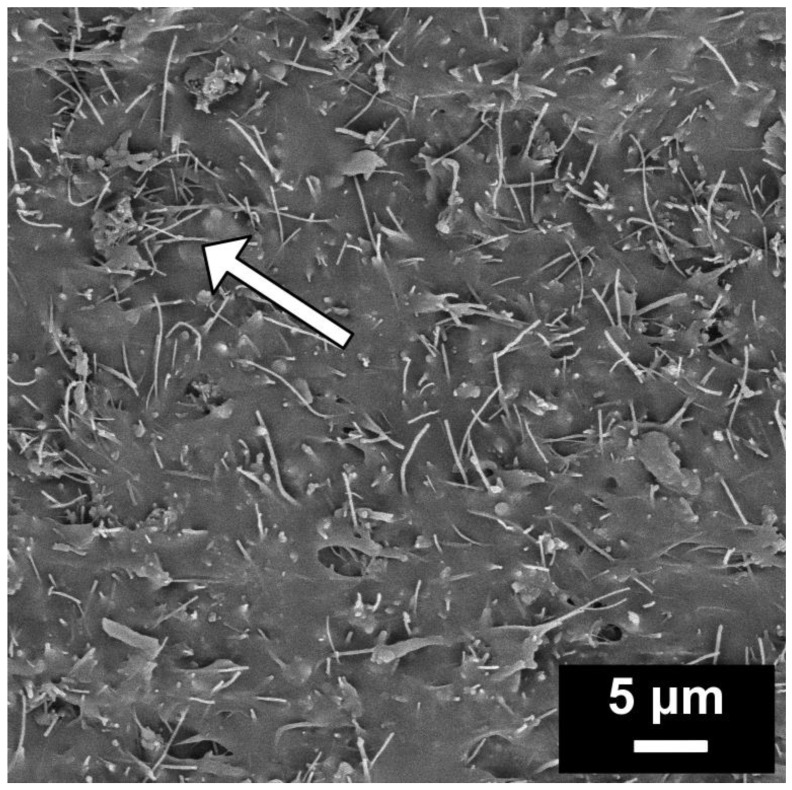
SEM image of EOC/CF composite with 30 wt.% of CF.

**Figure 2 polymers-15-02005-f002:**
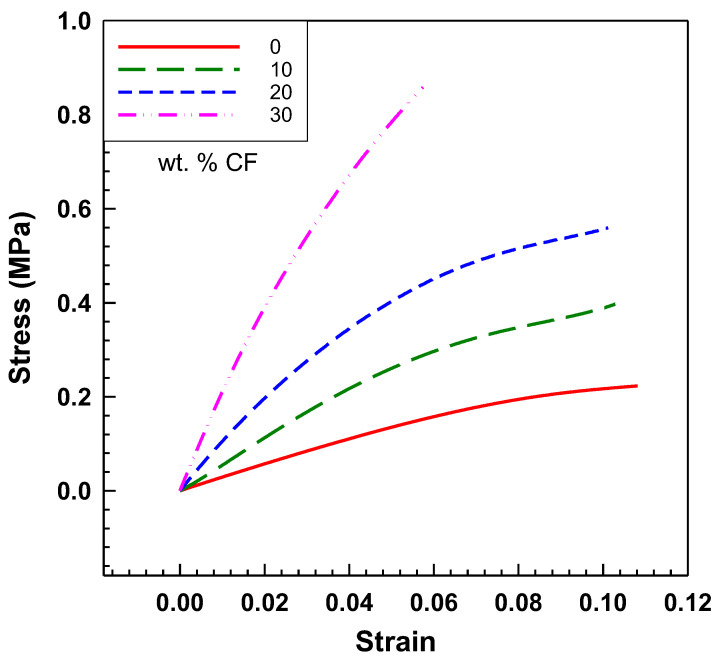
Tensile stress–strain curves of EOC/CF composites measured by DMA at room temperature.

**Figure 3 polymers-15-02005-f003:**
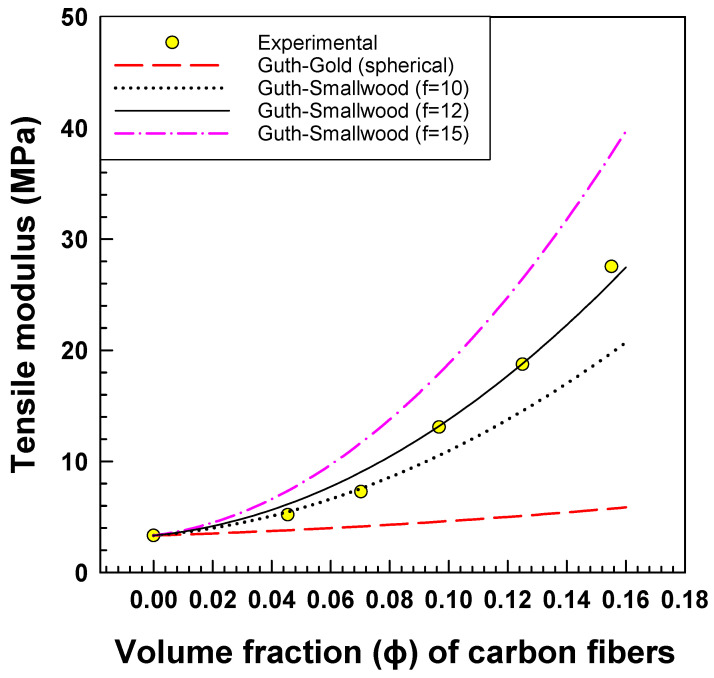
Experimental tensile modulus vs. prediction of tensile modulus by Guth–Gold model for spherical and Guth–Smallwood model for non-spherical particles.

**Figure 4 polymers-15-02005-f004:**
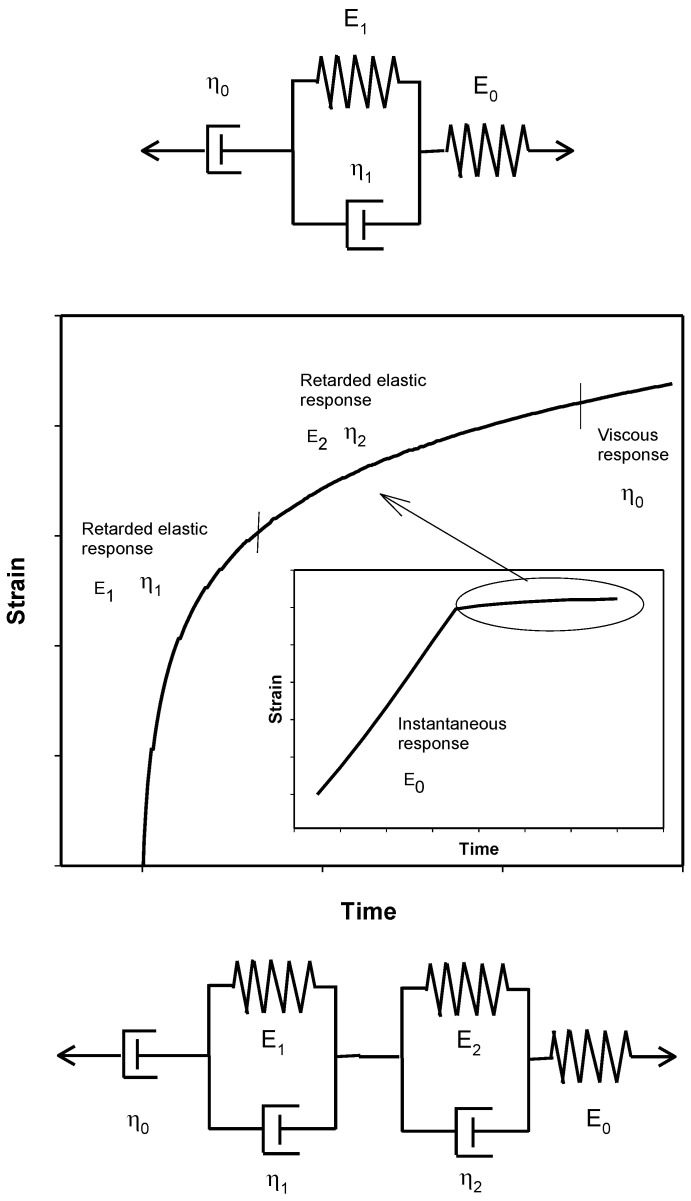
Scheme of Burgers model (four-parameter) and six-parameter model.

**Figure 5 polymers-15-02005-f005:**
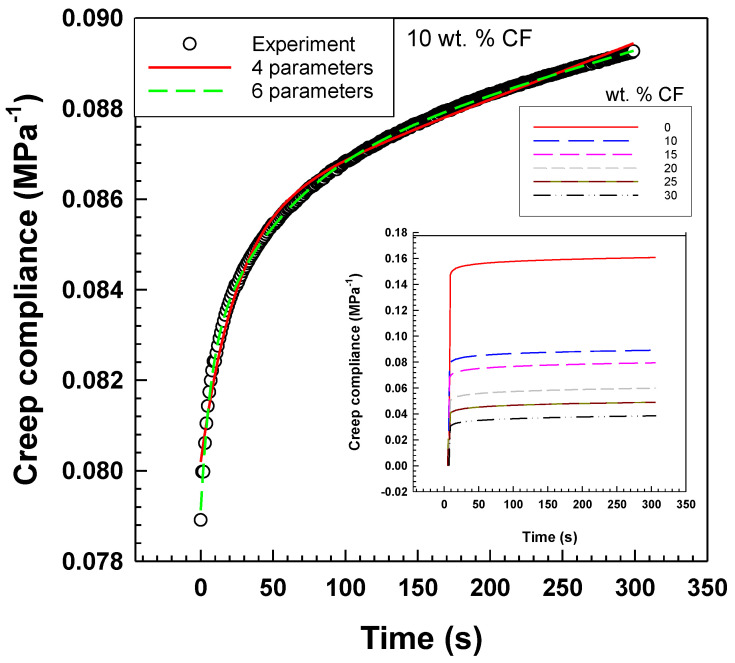
Creep compliance curves of EOC/CF composites measured by DMA at room temperature. Experimental data vs. Burgers model and six-parameter model.

**Figure 6 polymers-15-02005-f006:**
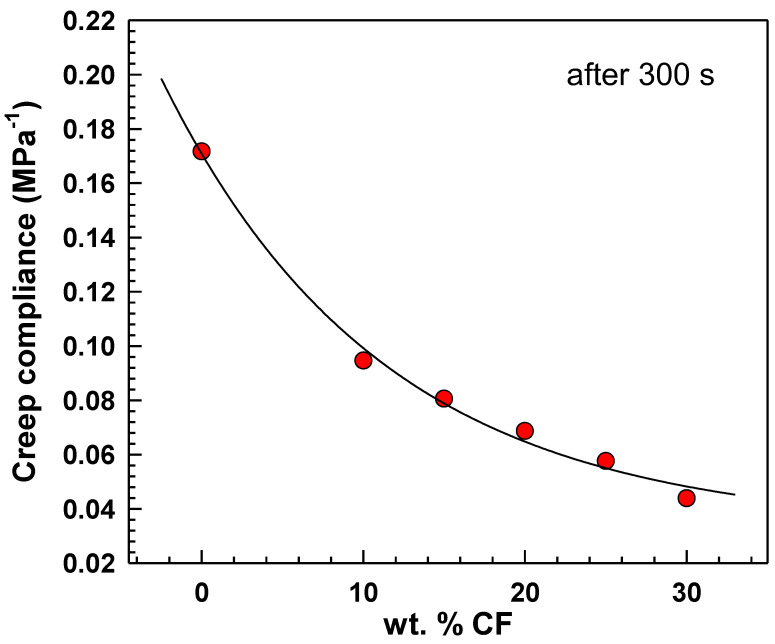
Creep compliance vs. CF content. Evaluated after 300 s from start of the measurement.

**Figure 7 polymers-15-02005-f007:**
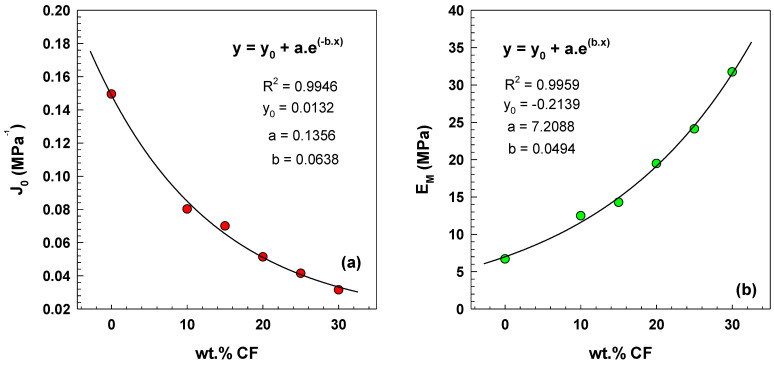
(**a**) Parameter *J*_0_ vs. CF content; (**b**) parameter *E_M_* vs. CF content.

**Figure 8 polymers-15-02005-f008:**
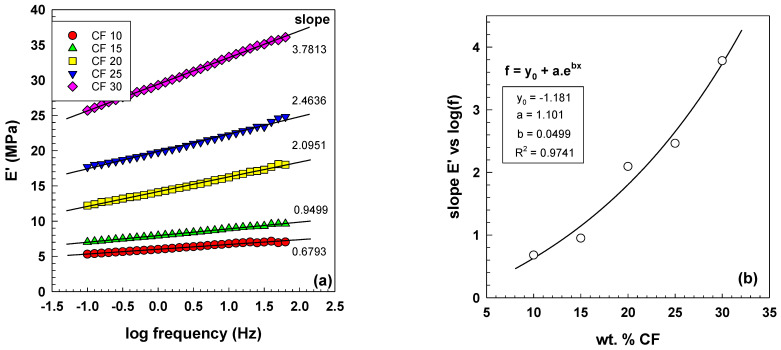
(**a**) Storage modulus as a function of frequency for EOC/CF composites measured by DMA at room temperature; (**b**) dependence of slope E’ vs. log(f) on CF content.

**Figure 9 polymers-15-02005-f009:**
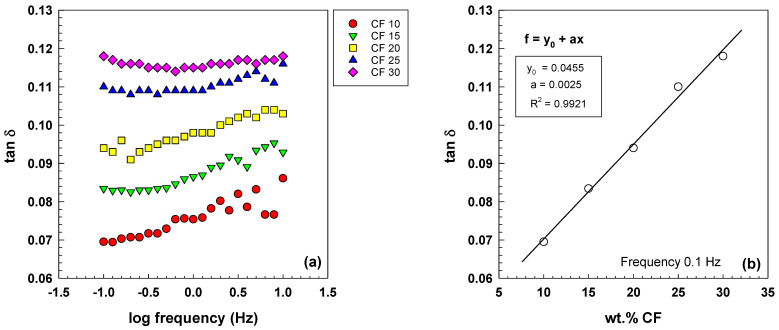
(**a**) Tan δ as a function of frequency for EOC/CF composites measured by DMA at room temperature; (**b**) dependence of Tan δ on CF content for frequency 0.1 Hz.

**Figure 10 polymers-15-02005-f010:**
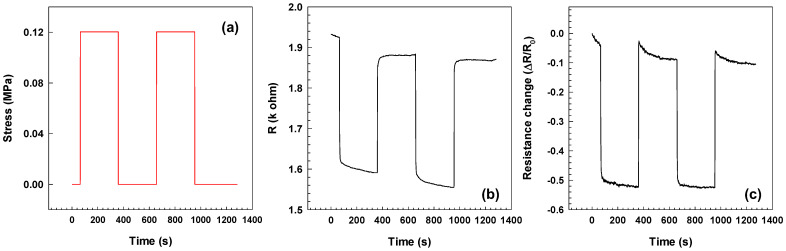
Loading/unloading cycles for (**a**) stress, (**b**) resistivity and (**c**) resistance change for EOC composite with 25 wt.% CF.

**Figure 11 polymers-15-02005-f011:**
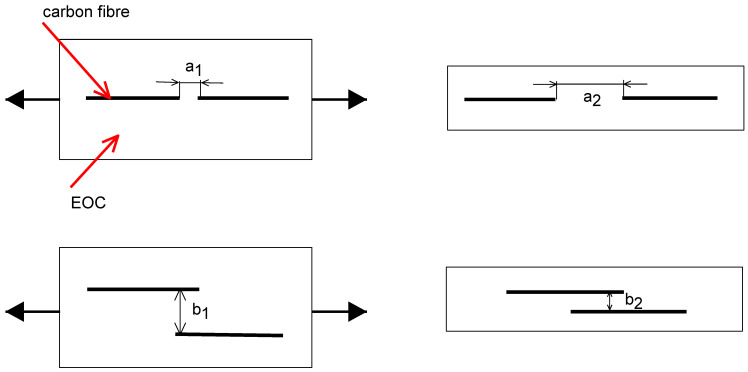
Schema of conductivity path explaining negative resistance change and gauge factor.

**Figure 12 polymers-15-02005-f012:**
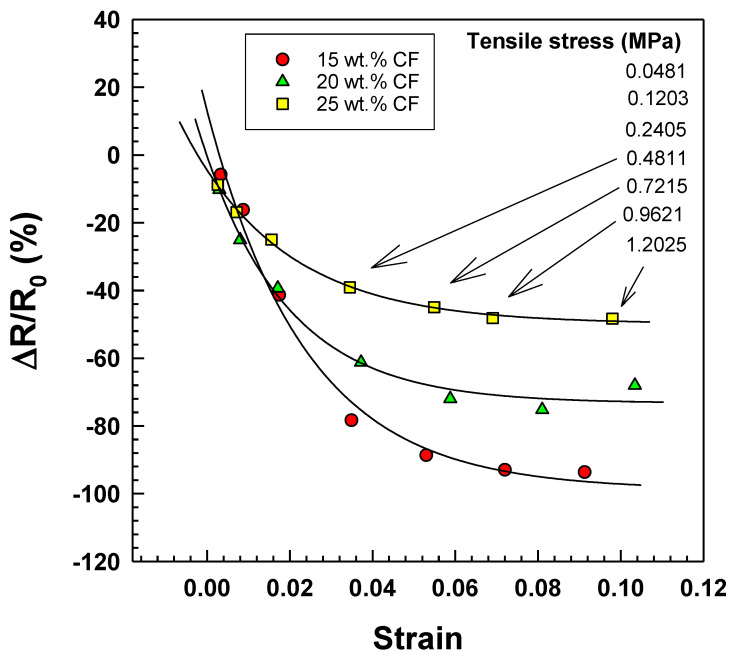
Strain dependence of resistance change for EOC/CF composites with various tensile stresses.

**Figure 13 polymers-15-02005-f013:**
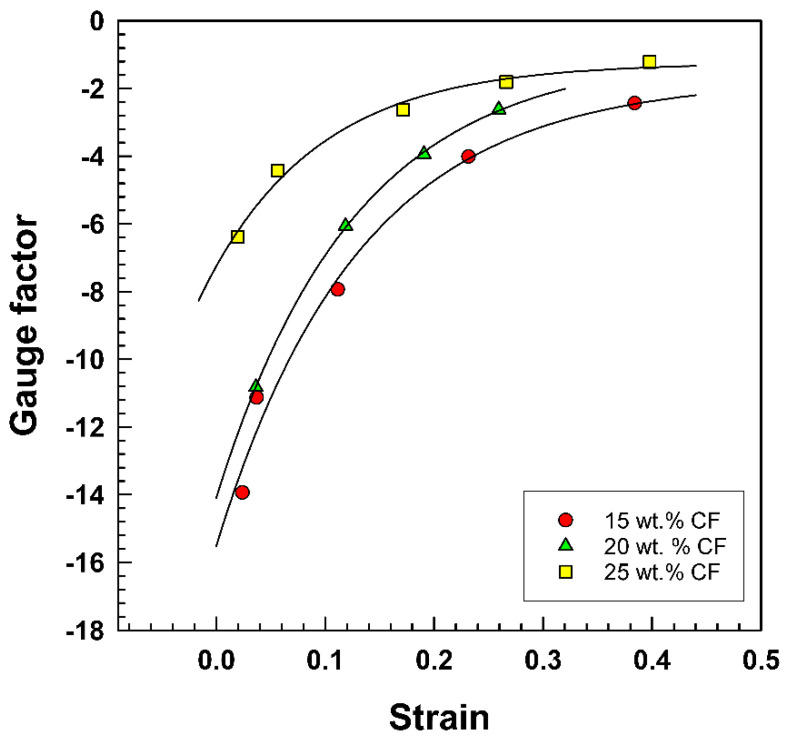
Strain dependence of gauge factor for EOC/CF composites.

**Figure 14 polymers-15-02005-f014:**
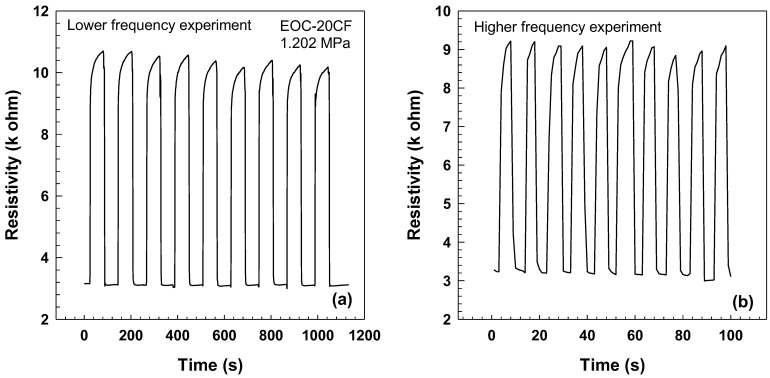
Resistivity vs. time for loading/unloading cycles after (**a**) 1 min and (**b**) 5 s of EOC composite with 20 wt.% CF and tensile stress 1.202 MPa.

**Table 1 polymers-15-02005-t001:** Properties of CF.

Average diameter	150 nm
Average length	15 μm
Aspect ratio	10–500
Density	2.0 g/cm^3^

**Table 2 polymers-15-02005-t002:** Values of the four-parameter model.

Sample	Four-Parameter Model				
	*E_m_* (MPa)	*E_k_* (MPa)	*η_m_* (MPa·s)	*η_k_* (MPa·s)	R^2^
EOC	6.7	142.6	67,114.7	3099.2	0.991
EOC/10 wt% CF	12.5	181.6	79,980.1	4501.2	0.993
EOC/15 wt% CF	14.3	175.1	76,050.2	4321.9	0.993
EOC/20 wt% CF	19.5	193.3	83,418.2	5017.1	0.992
EOC/25 wt% CF	24.1	222.2	92,440.7	6160.3	0.994
EOC/30 wt% CF	31.7	240.1	97,204.5	6803.7	0.994

**Table 3 polymers-15-02005-t003:** Values of the six-parameter model.

Sample	Six-Parameter Model						
	*E*_0_ (MPa)	*η*_0_ (MPa·s)	*E*_1_ (MPa)	*η*_1_ (MPa.s)	*E*_2_ (MPa)	*η*_2_ (MPa·s)	R^2^
EOC	6.8	86,936.1	220.2	978.6	194.9	8612.1	0.9995
EOC/10 wt% CF	12.6	109,715.6	298.2	1964.3	244.6	13,323.8	0.9996
EOC/15 wt% CF	14.5	102,613.3	286.7	1684.1	231.4	12,130.8	0.9995
EOC/20 wt% CF	20.1	108,270.9	327.2	1480.5	239.9	11,713.2	0.9996
EOC/25 wt% CF	24.7	120,924.9	403.9	2206.1	271.9	14,217.2	0.9997
EOC/30 wt% CF	32.8	123,687.4	437.8	2034.5	286.3	14,497.8	0.9996

**Table 4 polymers-15-02005-t004:** Values of negative resistance change.

wt.% CF	y_0_	a	b	R^2^
15	−98.82	111.8	41.96	0.9886
20	−73.32	72.92	48.76	0.9876
25	−49.85	45.36	41.06	0.9978

**Table 5 polymers-15-02005-t005:** Values of negative gauge factor.

wt.% CF	y_0_	a	b	R^2^
15	−15.52	13.81	7.619	0.9824
20	−14.09	13.13	7.902	0.9841
25	−7.252	6.008	9.577	0.9881

## Data Availability

The data presented in this study are available on request from the corresponding author.

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
