# Peer review of "Elastic Electrically Conductive Composites Based on Vapor-Grown Carbon Fibers for Use in Sensors"

_polymers, 2023, doi:10.3390/polym15092005_

Round 1
Reviewer 1 Report
The submitted manuscript reported the composite with VGCF (vapor grown carbon fiber), but their paper title only showed CF (carbon fiber). Since there are many carbon fibers being produced including carbon nano-fibers, it seems not clear to readers if the authors only use CF. Most CF are produced from the precursor fiber (oxidized fiber) and VGCF are different from them. The authors should make them more clear to readers.
Author Response
Dear Reviewer # 1:
I would like to express my sincere gratitude for taking the time to review our research article. Your feedback and recommendations have been extremely useful in improving the quality of our work.
Reviewer #1:
The submitted manuscript reported the composite with VGCF (vapor grown carbon fiber), but their paper title only showed CF (carbon fiber). Since there are many carbon fibers being produced including carbon nanofibers, it seems not clear to readers if the authors only use CF. Most CF are produced from the precursor fiber (oxidized fiber) and VGCF are different from them. The authors should make them clearer to readers.
- The paper title is updated to:
“Elastic electrically conductive composites based on vapor-grown carbon fibers for use in sensors”
Reviewer 2 Report
Authors study the effect of CF wt% on mechanical and morphology of EOC/CF composite synthesized using ultrsonication. They have provided experimentally gathered strain-stress curves and compared them to analytical methods. They show that addition of CF improved the tensile modulus of the EOC/CF composites. They also studied the electrical properties of the composites and found that the percolation threshold happens at 10 CF wr%.
The manuscripts (MS) is very well written and all the relevant references have been cited. The results are explained thoroughly and clearly. This MS can be accepted with minor editorial changes.
Author Response
Dear Reviewer # 2:
I would like to express my sincere gratitude for taking the time to review our research article. Your feedback and recommendations have been extremely useful in improving the quality of our work.
Reviewer #2:
The manuscripts (MS) is very well written and all the relevant references have been cited. The results are explained thoroughly and clearly. This MS can be accepted with minor editorial changes.
- The editorial comments are answered and the manuscript is updated.
Editor Comments:
1) The commas in Tables 2 and 4 should be changed to decimal points (.)
- Tables are corrected.
2) In figure 4, correct "reponse" to "response"
- Figure 4 is corrected
3) Figures 7, 8, 9, 10, 14 should be shown side by side, and not vertically (the authors should check the journal template for presenting figures).
- All vertical figures are updated to be side by side.
Reviewer 3 Report
Paper on mechanical and electrical properties of composites. I have the following comments:
English and typographical errors e.g. “The Morphology …” but there are others.
“Stress increases with higher content of the filler. “ - for same strain or strength - please make clear - this is also a general comment and so the English can be improved to make clear statements.
Tables of results - avoid use of comma “,” in data and use “.” e.g. 6.7 and not 6,7
No need for figures to be so large. e.g. Fig 2.
Figure 3 “Fraction, f, of carbon fibres” - this is better
Figure 5 , same colours style used for wt.% and parameter e.g. dark blue dashed line - please use different style.
Figure 13 - what is purpose of green shaded area?
Carbon-fibre based sensors examine here also, where the AC conductivity is also on interest to understand fibre distribution.
Soft Controllable Carbon Fibre-based Piezoresistive Self-Sensing Actuators
M Pan et al
Actuators 9 (3), 79 2020
Carbon fibre based flexible piezoresistive composites to empower inherent sensing capabilities for soft actuators
X Yan et al
Soft matter 15 (40), 8001-8011 2019
Author Response
Dear Reviewer # 3:
I would like to express my sincere gratitude for taking the time to review our research article. Your feedback and recommendations have been extremely useful in improving the quality of our work.
Reviewer #3:
Paper on mechanical and electrical properties of composites. I have the following comments:
1) English and typographical errors e.g. “The Morphology …” but there are others.
- It is corrected. Also, some grammar errors are corrected.
2) “Stress increases with higher content of the filler. “ - for same strain or strength - please make clear - this is also a general comment and so the English can be improved to make clear statements.
- line 107-108 „Stress increases with higher content of the filler for the same strain implying higher modulus. “
3) Tables of results - avoid use of comma “,” in data and use “.” e.g. 6.7 and not 6,7
- Tables are corrected.
4) No need for figures to be so large. e.g. Fig 2.
- Figures are resized.
5) Figure 3 “Fraction, f, of carbon fibres” - this is better
- Figure 3 is updated.
6) Figure 5, same colours style used for wt.% and parameter e.g. dark blue dashed line - please use different style.
- Figure 5 is updated.
7) Figure 13 - what is purpose of green shaded area?
- It is a technical issue when it was saved as a PDF file, it is corrected.
8) Carbon-fibre based sensors examine here also, where the AC conductivity is also on interest to understand fibre distribution.
Line 247: „In our previous paper we focused on AC conductivity [14] while this work is focused on DC conductivity change with stretching. “
9) Soft Controllable Carbon Fibre-based Piezoresistive Self-Sensing Actuators
Pan M, et. al., Actuators 9 (3), 79 2020
- Added to the manuscript [37].
10) Carbon fibre based flexible piezoresistive composites to empower inherent sensing capabilities for soft actuators
X Yan et. al., Soft matter 15 (40), 8001-8011 2019
- Added to the manuscript [36].